# Diclofenac sensitizes multi-drug resistant *Acinetobacter baumannii* to colistin

**Fabiana Bisaro[1], Clay D. Jackson-Litteken[1], Jenna C. McGuffey[1], Anna J. Hooppaw[1], Sophie Bodrog[2], Leila Jebeli[3], Manon Janet-Maitre[1], Juan C. Ortiz-Marquez[2,4], Tim van Opijnen[4], Nicholas E. Scott[3], Gisela Di Venanzio[1]\*, Mario F. Feldman[1]\***

**1** Department of Molecular Microbiology, Washington University School of Medicine in St. Louis; St. Louis, Missouri, United States of America, **2** Biology Department, Boston College; Chestnut Hill, Massachusetts, United States of America, **3** Department of Microbiology and Immunology, The Peter Doherty Institute for Infection and Immunity, University of Melbourne; Melbourne, Australia, **4** Boston Children's Hospital, Division of Infectious Diseases, Harvard Medical School, Boston, Massachusetts, United States of America

\* divenanzio.g@wustl.edu (GDV); mariofeldman@wustl.edu (MFF)

**Data Availability Statement:** The transcriptomics data has been deposited in SRA database with the data set identifier: PRJNA1109398 https://www.ncbi.nlm.nih.gov/sra/PRJNA1109398. The mass

## Abstract

*Acinetobacter baumannii* causes life-threatening infections that are becoming difficult to treat due to increasing rates of multi-drug resistance (MDR) among clinical isolates. This has led the World Health Organization and the CDC to categorize MDR *A. baumannii* as a top priority for the research and development of new antibiotics. Colistin is the last-resort antibiotic to treat carbapenem-resistant *A. baumannii*. Not surprisingly, reintroduction of colistin has resulted in the emergence of colistin-resistant strains. Diclofenac is a non-steroidal anti-inflammatory drug used to treat pain and inflammation associated with arthritis. In this work, we show that diclofenac sensitizes colistin-resistant *A. baumannii* clinical strains to colistin *in vitro* and in a murine model of pneumonia. Diclofenac also reduced the colistin minimal inhibitory concentration (MIC) of *Klebsiella pneumoniae* and *Pseudomonas aeruginosa* isolates. Transcriptomic and proteomic analyses revealed an upregulation of oxidative stress-related genes and downregulation of type IV pili induced by the combination treatment. Notably, the concentrations of colistin and diclofenac effective in the murine model were substantially lower than those determined *in vitro*, implying a stronger synergistic effect *in vivo* compared to *in vitro*. A *pilA* mutant strain, lacking the primary component of the type IV pili, became sensitive to colistin in the absence of diclofenac. This suggest that the downregulation of type IV pili is key for the synergistic activity of these drugs *in vivo* and indicates that colistin and diclofenac exert an anti-virulence effect. Together, these results suggest that diclofenac can be repurposed with colistin to treat MDR *A. baumannii*.

## Author summary

*Acinetobacter baumannii* causes infections that are difficult to treat due to high rates of antibiotic resistance, leading the World Health Organization and CDC to classify this pathogen as a top priority for research and development of new antibiotics. Colistin is one of few drugs still able to treat *A. baumannii* infections, but, recently, resistant strains have

spectrometry proteomics data including the full search results have been deposited in the Proteome Xchange Consortium via the PRIDE partner repository with the data set identifier: PXD047586.

**Funding:** This work was supported by the National Institutes of Health grants R01AI166359 to MFF and R01AI148470 to TVO and JOM; the Australian Research Council Future Fellowship FT200100270 to NES; the Discovery Project Grant DP210100362 to NES, and the Australian National Health and Medical Research Council Ideas grant 2018980 to NES. The funders had no role in study design, data collection and analysis, decision to publish, or preparation of the manuscript.

**Competing interests:** The authors have declared that no competing interests exist.

emerged. In this work, we show that a combination of diclofenac, an FDA-approved anti-inflammatory drug, and colistin can kill colistin-resistant *A. baumannii*. Concentrations of colistin and diclofenac effective in the context of an animal model were substantially lower than *in vitro*, implying a stronger synergistic effect in the animal. Further studies revealed an increase in oxidative stress and decrease in type IV pili with the combination treatment. In the animal model, colistin-resistant *A. baumannii* lacking type IV pili became sensitive to colistin alone, which suggests that the decrease of type IV pili is key for the synergistic activity of these drugs. Diclofenac in combination with colistin was also efficient in killing other multidrug-resistant pathogens as well, including *Klebsiella pneumoniae* and *Pseudomonas aeruginosa*. Together, our results suggest that diclofenac can be used in combination with colistin to treat multidrug-resistant *A. baumannii* infections.

## Introduction

*Acinetobacter baumannii* causes a wide range of life-threatening nosocomial infections, often in immunocompromised patients. Unfortunately, these infections are becoming more difficult to manage due to a sharp rise in multi-drug resistance rates among *Acinetobacter* species, which are now nearly four times higher than those observed in other Gram-negative pathogens [1]. One of the last resort antibiotics used to treat multi-drug resistant (MDR) isolates of *A. baumannii* is carbapenem. However, the rate of resistance to carbapenems in *A. baumannii* has now exceeded 40% in the U.S. [2]. Because of these increasing antibiotic resistance rates, the World Health Organization has ranked carbapenem-resistant *A. baumannii* as a top priority for research to develop new antimicrobial therapies [3]. Carbapenem-resistant strains are frequently resistant to all other agents except colistin [4].

Colistin (polymyxin E) is a nonribosomally synthesized cationic antimicrobial polypeptide that is part of the polymyxin class of antibiotics. Its bactericidal activity occurs when positively charged colistin interacts with the negatively charged lipid A moiety of lipopolysaccharide (LPS) in the outer membrane of Gram-negative bacteria [5]. This interaction leads to the displacement of $Ca^{2+}$ and $Mg^{2+}$ ions from the phosphate groups of LPS, causing destabilization of the membrane [5]. Although the direct mechanism of colistin killing is not well-understood, it is believed that disruption of both the outer and inner membranes leads to leaking of cytoplasmic contents and cell death [6]. Additional mechanisms of colistin-mediated killing have been described, such as formation of hydroxyl radicals as colistin molecules cross the membrane leading to production of reactive oxygen species (ROS) that damage the bacterial cell [7]. Colistin is considered the last-resort antibiotic for treating severe infections caused by carbapenem-resistant *A. baumannii* [8–13] due to its associated toxicity [14–22]. Not surprisingly, reintroduction of colistin to treat carbapenem-resistant infections has resulted in the emergence of colistin-resistant *A. baumannii* [2,23,24]. In *A. baumannii*, several mechanisms of colistin resistance have been identified including the alteration of the lipid A component of the LPS by incorporating a phosphoethanolamine moiety, that reduces the overall negative charge of the LPS. This is mediated by the chromosomal-encoded phosphoetanolamine transferases, PmrC or EptA, as well as plasmid-encoded mobile colistin resistant genes (*mcr*) [25]. Interestingly, some colistin-resistant *A. baumannii* strains have even evolved to completely lose LPS by inactivation of genes in its biosynthetic pathway, such as *lpxA*, *lpxC*, *lpxD* [26]. Another relevant mechanism involves the efflux of colistin from the cell mediated by the EmrAB system [27–29]. Thus, research into development of therapeutics or alternative mechanisms to treat carbapenem- and colistin-resistant *A. baumannii* is critical.

In a previous study, we described a novel stress response mechanism in *A. baumannii*, involving the phenylacetic acid (PAA) pathway. We found that in presence of subinhibitory concentrations of antibiotics and other stressors, *Acinetobacter* modulates the expression of the *paa* operon. This operon encodes enzymes responsible for the degradation of PAA [30]. Mutations in the *paa* operon results in increased susceptibility to antibiotics. These findings suggest that accumulation of PAA is detrimental for bacterial growth and that interfering with PAA levels disrupts the programmed antibiotic-mediated adaptations of *A. baumannii*. Therefore, we hypothesized that the exogenous addition of PAA derivatives may have therapeutic potential. Diclofenac is as non-steroidal anti-inflammatory drug (NSAID) that belongs to a family of PAA derivatives approved by the U.S. Food and Drug Administration (FDA) for use in humans. Diclofenac functions by inhibiting cyclooxygenases, COX-1 and COX-2 in the host [31]. It has been suggested that diclofenac may have antibiotic properties [32–35]. In this study, we explore the potential of repurposing the FDA approved PAA-derivative, diclofenac, to enhance antibiotic efficacy against MDR *A. baumannii* with a specific focus on the last-resort drug, colistin.

## Results

### Diclofenac increases colistin susceptibility in clinical *A. baumannii* isolates

We previously observed that exogenous addition of PAA or inactivation of the *paa* operon increases the sensitivity of *A. baumannii* to multiple antibiotics [30]. Therefore, we hypothesized that FDA-approved PAA derivatives, such a diclofenac, could also interfere with the bacterial response to antibiotics. As MDR *A. baumannii* is rapidly losing susceptibility to colistin, we first investigated whether FDA-approved PAA derivatives could increase susceptibility to this last line of defense antibiotic. To test this, we used *A. baumannii* ARC6851 [36], a strain encoding several genes related to colistin resistance as *pmrC* (OB946_RS03455), *pmrAB* (OB946_RS03460-5) and the efflux system, *emrAB* (OB946_RS14565-70), and exhibiting high resistance to colistin with a minimum inhibitory concentration (MIC) >1024 µg/ml. We treated ARC6851 with a sub-MIC concentration of colistin (256 µg/ml) in combination with 100 µM phenylacetic acid, 4-fluorophenylacetic acid, 4-hydroxyphenylacetic acid, mandelic acid, ibufenac, ibuprofen, or diclofenac. We found that colistin, in conjunction with diclofenac but no other PAA derivatives, inhibits the growth of ARC6851 (**Fig 1**).

To determine whether the enhancement of colistin susceptibility by diclofenac was specific to ARC6851, we assessed the MIC of several *A. baumannii* strains with varied susceptibilities to colistin. Additionally, we assessed other prominent Gram-negative pathogens, including *Klebsiella pneumoniae*, *Pseudomonas aeruginosa*, and *Enterobacter cloacae*. This analysis was conducted using a 2-fold broth dilution microtiter assay (**S1 Table**). We observed that diclofenac increased the susceptibility of all six *A. baumannii* strains tested. The most significant change in MIC, with an increase of ≥4-fold, occurred in the colistin-resistant strains AB347, AB431, AB774 and ARC6851. We also found that diclofenac renders *K. pneumoniae*, *E. cloacae* and *P. aeruginosa* more susceptible to colistin (**S1 Table**). Hence, the susceptibility to colistin mediated by diclofenac is not exclusive to *A. baumannii*. While colistin does not typically target Gram-positive bacteria, we assessed the MIC for *Staphylococcus aureus*. However, we did not observe any changes in the MIC in the presence of diclofenac.

Treatment of *A. baumannii* UPAB1, AB431, ARC6851 and AB347 with either 100 µM diclofenac or the solvent control DMSO, did not affect *Acinetobacter* growth in the absence of colistin (**Fig 2A, left panel**). However, when adding sub-MIC concentrations of colistin, diclofenac inhibited the growth of the four strains tested (**Fig 2A, right panel**). Similar results were obtained for a panel of *K. pneumoniae* (**Fig 2B**) and *P. aeruginosa* (**S1 Fig**) strains.

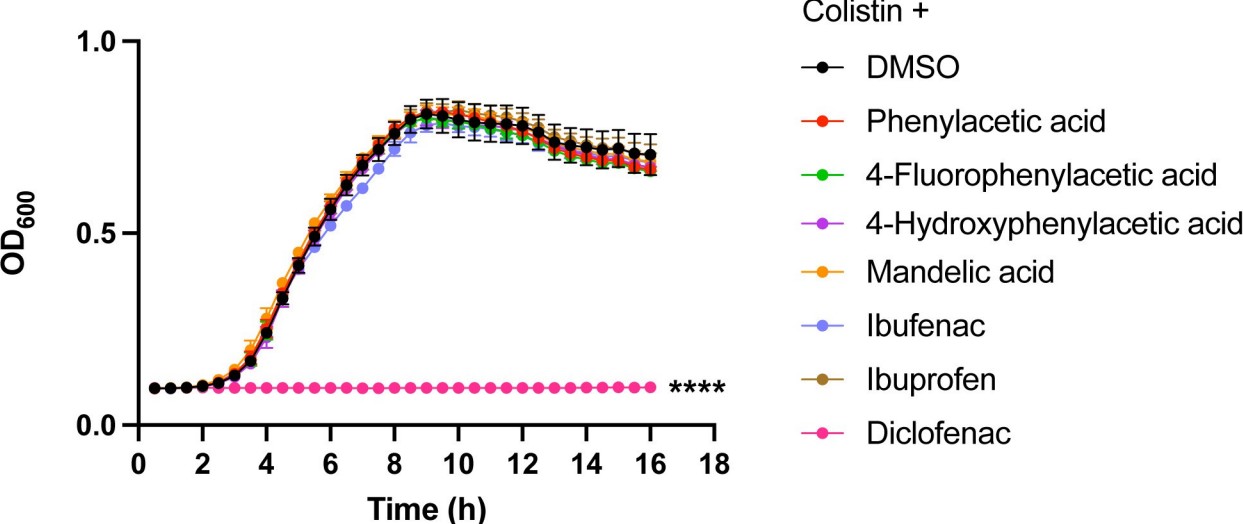

**Fig 1. Diclofenac in combination with colistin inhibits *A. baumannii* growth.** Representative growth curves of ARC6851 in LB containing 256 µg/ml colistin and either the solvent control DMSO or 100 µM phenylacetic acid, 4-fluorophenylacitic acid, 4-hydroxyphenylacetic acid, mandelic acid, ibufenac, ibuprofen, or diclofenac. Data is representative of three biological replicates. ****$P<0.0001$ (One-way ANOVA with Tukey's test for multiple comparisons, results were compared to the control group treated with DMSO).

## Colistin, but not diclofenac, increases the permeability of ARC6851

Colistin targets LPS in the outer membrane [5,7]. On the other hand, it has been reported that large concentrations of diclofenac inhibits bacterial DNA synthesis [37,38]. Therefore, we hypothesized that the growth inhibition caused by colistin plus diclofenac may result from the membrane-permeabilizing activity of colistin. To test this hypothesis, we measured the membrane permeability of our most colistin-resistant isolate, ARC6851, under treatment with colistin alone, diclofenac alone, and in combination. Permeability was analyzed by measuring the accumulation of the fluorescent dye Hoechst H33342, which binds to DNA, in treated cells [39,40]. As anticipated, colistin increased bacterial permeability in a dose-dependent manner (**Fig 3A**). Notably, we found that 100 µM diclofenac (the concentration employed in our MIC assays) did not alter the permeability of ARC6851 when used alone (**Fig 3A**) or in combination with different concentrations of colistin (**S2A Fig**). Moreover, similar results were obtained when measuring cell permeability with N-phenyl-1-naphthylamine (NPN). The addition of the efflux pump inhibitor Carbonyl cyanide-m-chlorophenylhydrazone (CCCP) did not modify cell permeability (**S2B Fig**). These results indicate that colistin, but not diclofenac, affects membrane permeability in ARC6851. This result is in agreement with previous work in which colistin increased the permeability of bacteria to other molecules [41].

## Bacterial cell permeability is not sufficient for diclofenac-mediated killing

The effect of colistin on permeability prompted us to question whether colistin merely acts as a permeabilizing agent to enable the diclofenac-mediated elimination of *A. baumannii*. To test this possibility, we performed growth curves of ARC6851 treated with 100 µM diclofenac with the combination of a different permeabilizing agent, consisting of 0.01% sodium dodecyl sulfate (SDS) and 0.075 mM ethylenediaminetetraacetic acid (EDTA) (**Fig 3B**) [42]. As a control, we included ARC6851 treated with vancomycin, a large compound targeting the cell wall that typically is only used to treat Gram-positive bacterial infections [43, 44]. However, previous reports have shown that increasing the cell permeability of Gram-negative bacteria can lead to

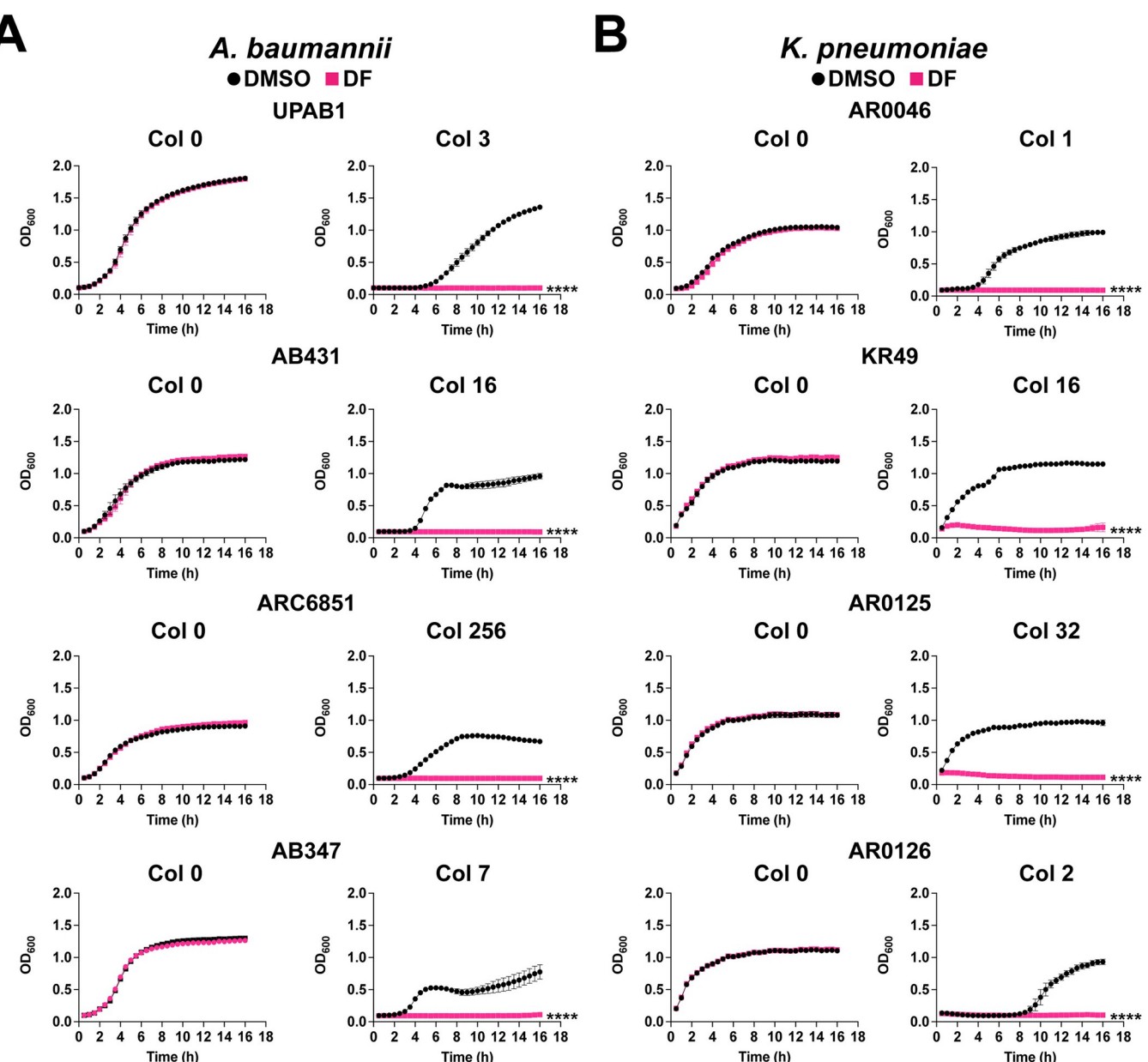

**Fig 2. Diclofenac inhibits *A. baumannii* and *K. pneumoniae* growth in the presence of colistin.** (**A**) Representative growth curves of *A. baumannii* UPAB1, AB431, ARC6851 and AB347 strains in LB containing DMSO or 100 μM diclofenac (DF) (left panel) and 3 μg/ml, 16 μg/ml, 256 μg/ml, or 7 μg/ml colistin respectively (right panel). (**B**) Representative growth curves of *K. pneumoniae* AR0046, KR49, AR0125, AR0126 strains in LB containing DMSO or 100 μM diclofenac (DF) (left panel) and 1 μg/ml, 16 μg/ml, 32 μg/ml, or 2 μg/ml colistin respectively (right panel). Data is representative of three biological replicates. ****$P<0.0001$, unpaired *t* tests at 16 h for Col + DF 100 compared to DMSO control. Col (colistin), DF (diclofenac).

susceptibility to vancomycin [43,44]. Interestingly, although vancomycin in combination with SDS/EDTA increased killing of ARC6851, 100 μM diclofenac in combination with SDS/EDTA treatment did not affect ARC6851 growth (**Fig 3B**). Polymyxin B Nonapeptide (PBNP) is a derivative that lacks a fatty acid tail and is significantly less active than colistin and polymixin B against species such as *E. coli* and *K. pneumonia*, but still possesses outer membrane-permeabilizing activity. PBNP did not show any activity against *A. baumannii* (**S3A Fig**). Together, our data demonstrate that indeed, both colistin and diclofenac are required to kill *A. baumannii*.

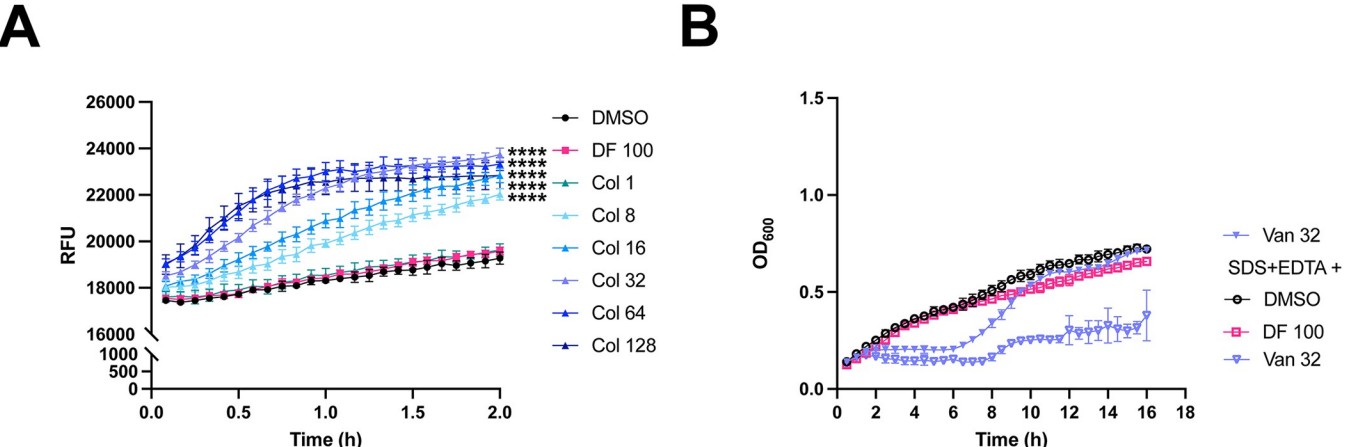

**Fig 3. Colistin, but not diclofenac, increases membrane permeability of ARC6851. (A)** Membrane permeability was assessed by measuring uptake of Hoescht 33342 (H33342) fluorescent dye. Bacterial cells were suspended in PBS supplemented with DMSO, 100 µM diclofenac (DF) or increasing concentrations of colistin (Col 1 µg/ml, 8 µg/ml, 16 µg/ml, 32 µg/ml, 64 µg/ml, or 128 µg/ml). Relative fluorescent units (RFU) were monitored over the course of 2 hours. **(B)** Representative growth curves of ARC6851 in LB containing 32 µg/ml Vancomycin or 0.01% SDS + 0.075 mM EDTA in combination with the solvent control DMSO, 100 µM Diclofenac or 32 µg/ml Vancomycin. Data is representative of three biological replicates, ****$P<0.0001$, (One-way ANOVA with Tukey's test for multiple comparisons, results were compared to the control group treated with DMSO). Col (colistin), DF (diclofenac), Van (Vancomycin), SDS (sodium dodecyl sulfate), EDTA (ethylenediaminetetraacetic acid).

We then investigated the combinatorial potential of diclofenac with colistin in a checkerboard assay of ARC6851 (**Fig 4A**). Colistin alone only killed ARC6851 at a concentration higher than 1024 µg/ml. Diclofenac alone did not have any effect on growth of ARC6851, even at 4,000 µM concentration (**Figs 4A and S3B**). In contrast, when we evaluated the combination of colistin plus diclofenac, we found growth inhibition at 256 µg/ml colistin plus 100 µM diclofenac (**Fig 4A**). Furthermore, the effect on *Acinetobacter* growth was dependent on both colistin and diclofenac concentrations (**Fig 4B and 4C**). These results indicate that diclofenac and colistin exert synergistic effects against *A. baumanni*.

It was previously reported that combinations of colistin and fatty acid synthesis (FAS), inhibitors such as triclosan, a FabI inhibitor, synergized with colistin against *mcr-1*-expressing *E. coli* [41] and that the phenotype was rescued by adding fatty acids [41]. Indeed, ARC6851 colistin MIC was reduced in the presence of triclosan and addition of oleic acid or linoleic acid abolished the synergy (**S2 Table**). However, the addition of palmitic acid, araquidonic acid, linoleic acid, oleic acid and stearic acid had no effect on *Acinetobacter* colistin MIC in the presence of diclofenac (**S3 Table**). These results indicate that the synergistic effects mediated by diclofenac and FAS inhibitors with colistin occur through different molecular mechanisms. Additionally, we found that the addition of diclofenac did not significantly alter the MICs of either ARC6851 or UPAB1 to other antibiotics such as, ciprofloxacin, erythromycin, tetracycline, vancomycin, imipenem and sulfomethoxazole/trimethoprim (**S4 Table**).

## Colistin plus diclofenac significantly alter ARC6851's gene and protein expression

To dissect the molecular mechanism underlying the synergistic effect of colistin plus diclofenac treatment, we performed whole-cell transcriptomic of ARC6851 grown in LB, LB plus colistin (1 µg/ml), LB plus diclofenac (100 µM), or LB plus colistin (1 µg/ml) and diclofenac (100 µM). We employed sub-MIC colistin concentrations that did not induce significant growth defects *in vitro* (**S4 Fig**). Only 4 genes involved in lipid metabolism and homeostasis

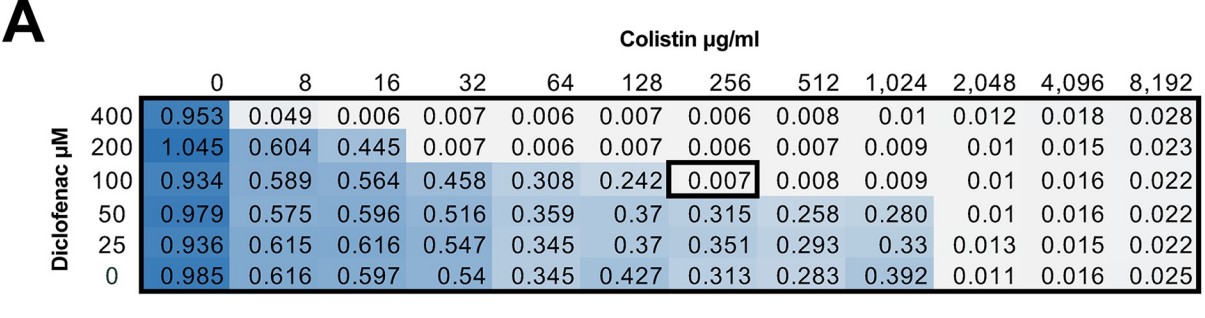

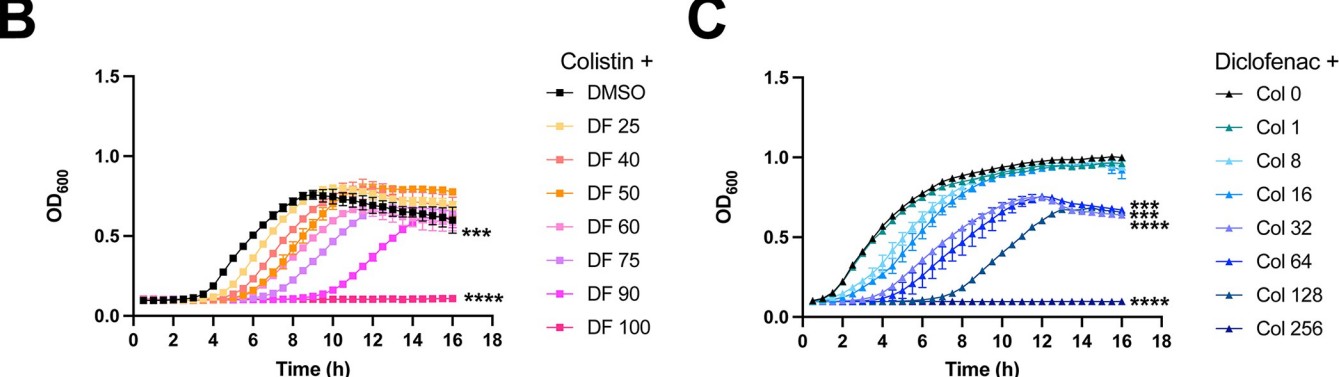

**Fig 4. Colistin and diclofenac inhibit growth of ARC6851 in a synergistic manner. (A)** Checkerboard microdilution assay for ARC6851 with colistin and diclofenac. Blue color represents cell density. **(B)** Growth curves of ARC6851 in LB containing 256 μg/ml colistin with either DMSO or increasing concentrations of diclofenac (25 μM, 40 μM, 50 μM, 60 μM, 75 μM, 90 μM, or 100 μM) **(C)** Growth curves of ARC6851 in LB containing 100 μM diclofenac with either DMSO or increasing concentrations of colistin (1 μg/ml, 8 μg/ml, 16 μg/ml, 32 μg/ml, 64 μg/ml, 128 μg/ml, or 256 μg/ml). Data is representative of three biological replicates ***P<0.001, ****P<0.0001 (One-way ANOVA with Tukey's test for multiple comparisons, results were compared to the control group treated with DMSO). Col (colistin), DF (diclofenac).

were upregulated in response to diclofenac alone, while no genes were downregulated in this condition (**Figs 5A** and **S5** and **S5 Table**). No statistically significant changes in gene expression were detected in response to colistin alone (**Fig 5A**). However, the combination treatment of colistin plus diclofenac led to differential expression of 130 genes, with 69 genes upregulated and 61 genes downregulated (**Fig 5A and 5B** and **S6 Table**). Remarkably, genes involved in the synthesis of type IV pili and catabolism of PAA were the most significantly repressed in presence of both drugs. The genes most upregulated by diclofenac and colistin are involved in oxidative stress response. Several efflux pumps were also induced. The complete functional analysis of the transcriptomic data using eggNOG Mapper [45,46] is shown in **S6 Fig**. As controls, we performed the comparisons between treatment groups colistin plus diclofenac against colistin (**S7 Table**) and colistin plus diclofenac against diclofenac (**S8 Table**) and found similar results than those of the colistin plus diclofenac against DMSO (**Fig 5A and 5B** and **S6 Table**). We further performed whole-cell differential proteomics of ARC6851 grown in the same experimental conditions (**Fig 5C and 5D** and **S9, S10** and **S11 Tables**). In agreement with the transcriptomic data, the most dramatic change detected by proteomics is the repression of type IV pili synthesis.

The transcriptomics and proteomics suggest that induction of oxidative stress and repression of pili expression are the two most important processes affected by diclofenac and colistin. Indeed, significantly fewer ARC6851 CFUs were recovered when exposed to 5mM $H_2O_2$ in the presence of colistin plus diclofenac compared to the untreated cells. (**Fig 6A**).

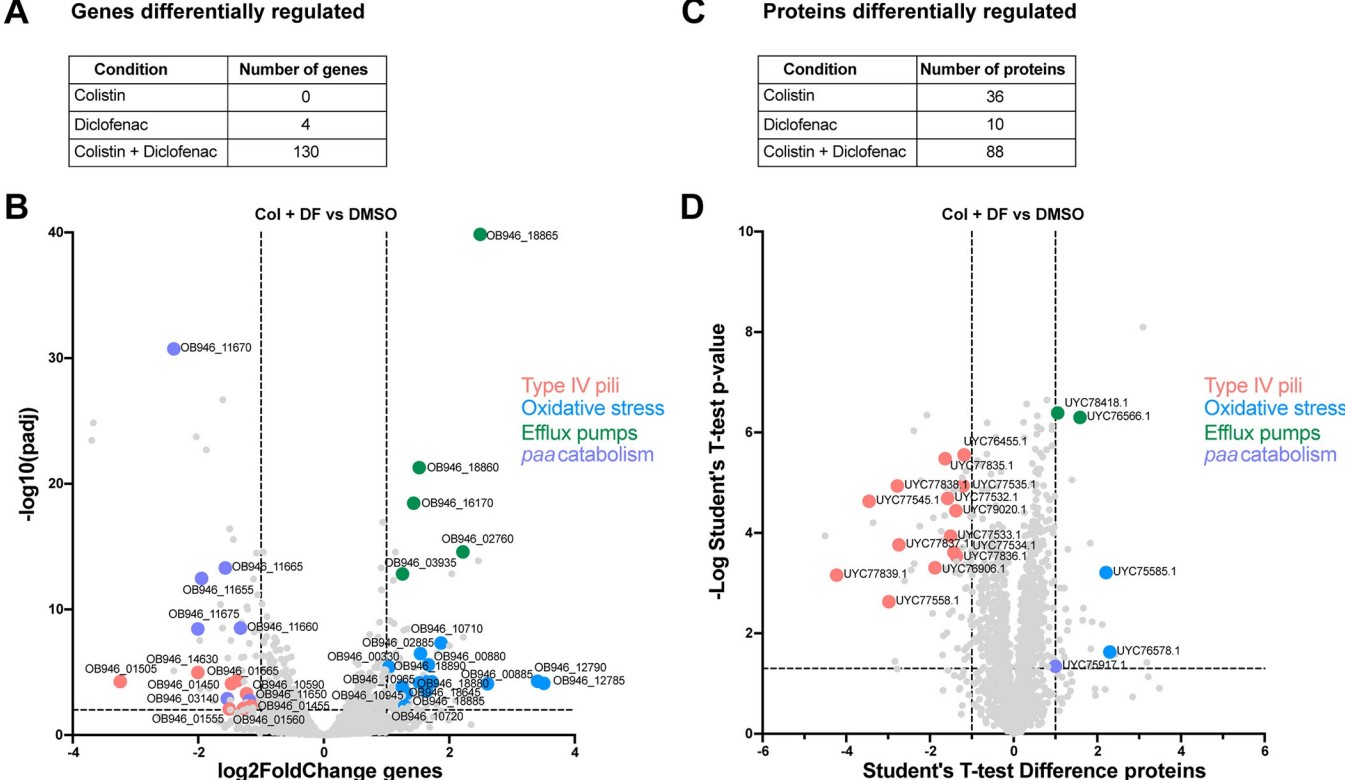

**Fig 5. Oxidative stress response and type IV pili are differentially regulated in ARC6851 under colistin and diclofenac combination treatment.** (A) Number of differentially regulated genes in ARC6851 when treated with 1 µg/ml colistin, 100 µM diclofenac, or the combination. (B) Volcano plot showing differentially expressed genes in ARC6851 with colistin and diclofenac combination treatment vs DMSO control. (C) Number of differentially regulated proteins in ARC6851 when treated with 1 µg/ml colistin, 100 µM diclofenac, or the combination. (D) Volcano plot showing differentially expressed proteins in ARC6851 in colistin plus diclofenac combination treatment vs DMSO control. Col (colistin), DF (diclofenac). Five biological replicates were prepared for each condition.

Furthermore, the presence of up to 1% DMSO, a ROS scavenging agent shown to reduce ROS-mediated killing in *E. coli* [47], improved ARC6851 growth in the presence of DF and colistin (**Fig 6B**). These results indicated that treatment of ARC6851 with both drugs renders bacteria susceptible to oxidative stress. Subsequently, we performed qRT-PCR to assess the expression of *pilA*, a crucial component of type IV pili, in ARC6851 and in the *A. baumannii* clinical isolate AB347 (**S7A Fig**). We observed a ~4 and ~3-fold downregulation of *pilA* respectively, when comparing the combination treatment of colistin plus diclofenac against DMSO, colistin, or diclofenac alone, consistent with the transcriptomic data (**Fig 5B**). Furthermore, a significant reduction in the quantity of type IV pili under the colistin plus diclofenac treatment was observed by transmission electron microscopy (**S7B and S7C Fig**).

## Colistin plus diclofenac is effective against colistin-resistant *A. baumannii* in vivo

Building on the encouraging *in vitro* findings, we conducted experiments to assess the efficacy of combining colistin and diclofenac for the treatment of *A. baumannii* pneumonia in an acute murine model of infection [36,48]. Briefly, mice were intranasally inoculated with ARC6851 and, at 4 h post-infection (hpi), mice were injected intraperitoneally (IP) with PBS, 2.5 mg/kg colistin, 1.25 mg/kg diclofenac, or 2.5 mg/kg colistin plus 1.25 mg/kg diclofenac.

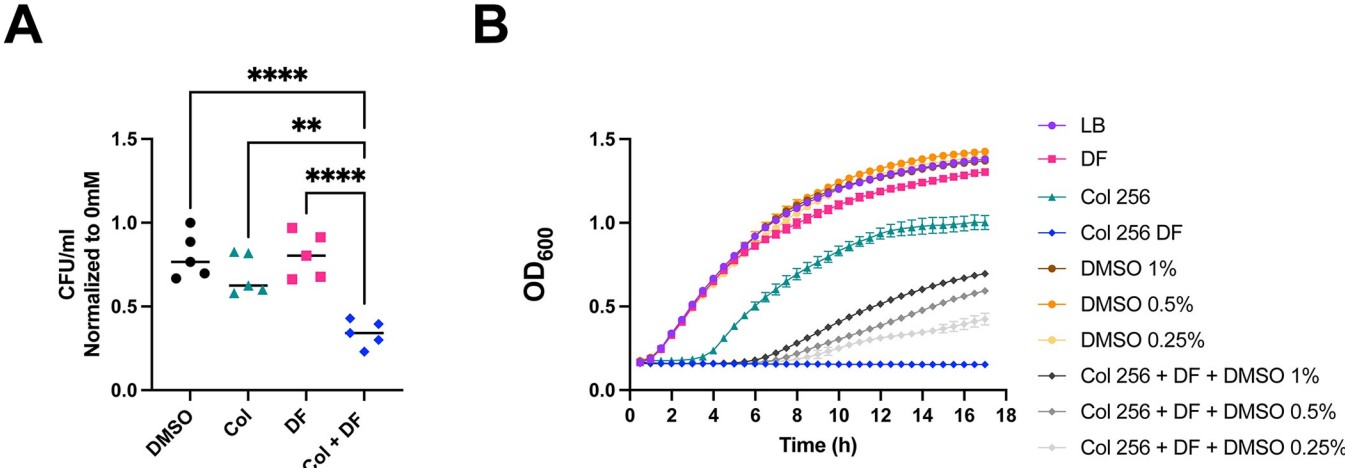

**Fig 6. Diclofenac and colistin–dependent bacterial killing is mediated by ROS. (A)** ARC6851 was grown to mid-exponential phase in LB + DMSO, LB + colistin (1 μg/ml) + DMSO, LB + diclofenac (100 μM), or LB + colistin (1 μg/ml) + diclofenac (100 μM), and treated with 0 mM or 5 mM $H_2O_2$ for 2 h. Survival was measured by serial dilution and quantification of recoverable CFU/mL. Symbols represent five biological replicates; the horizontal line represents the Median. **(B)** Representative growth curves of ARC6851 in LB containing 256 μg/ml colistin, 100 μM diclofenac and increasing concentrations of DMSO (0.25%, 0.5% and 1%). Data is representative of three biological replicates **$P<0.01$, ****$P< 0.0001$; One-way ANOVA, Tukey's test for multiple comparisons. Col (colistin), DF (diclofenac).

Colistin treatment was administered every 8 hours, starting at 4 hours post-infection (4, 12, and 20 hpi), in both the colistin-only and colistin plus diclofenac treatment groups. Diclofenac treatment was performed once at 4 hpi in the diclofenac-only and colistin plus diclofenac groups based on previously established murine treatment protocols [49–51]. At 24 hpi, the lungs, spleens and kidneys were harvested and processed to determine the bacterial burden. Individual treatments of colistin or diclofenac alone did not produce any significant changes in bacterial burden. However, the combination treatment of colistin plus diclofenac produced a significant reduction in recovered CFUs in lungs, kidneys, and spleens compared to all other groups (**Fig 7A**). The concentrations of diclofenac and colistin found to be effective in this model are 2.5 and 20 times lower, respectively, than those determined in MIC experiments. Moreover, no inhibition of cyclooxygenase COX-2 activity was observed *in vivo* under our model conditions (**S8 Fig**). This indicates that the synergistic effect of colistin plus diclofenac is significantly greater *in vivo* than *in vitro*. Furthermore, all mice treated with PBS or colistin alone succumbed to the infection within 36 hpi and 50.5 hpi, respectively (**Fig 7B**). In contrast, 40% of mice treated with the combination of colistin plus diclofenac survived up to 72 hpi. (end-point of the experiment). Notably, mice in the dual treatment group that survived the entirety of the survival study exhibited no symptoms associated with pneumonia (e.g., ruffled fur, hunching, dyspnea, reduced activity) by 72 hpi, consistent with resolution of disease.

### *In vivo*, type IV pili synergistic inhibition yields anti-virulence effects

Type IV pili is crucial for virulence in various pathogens such as *P. aeruginosa*, *V. cholerae*, and *Neisseria* spp., among others [52–56]. Transcriptomic and proteomic analyses revealed a drastic reduction in the expression of type IV pili-related genes in ARC6851 when exposed to the combined treatment of colistin and diclofenac. We hypothesized that the reduction of type IV pili by treatment with colistin and diclofenac enhances the bacterial clearance through an anti-virulence mechanism. To test this hypothesis, we constructed a Δ*pilA* strain. TEM analysis confirmed the absence of type IV pili in Δ*pilA* (**S9A Fig**). *In vitro*, Δ*pilA* MIC values and growth in the presence of colistin, diclofenac or the combination treatment was similar to the

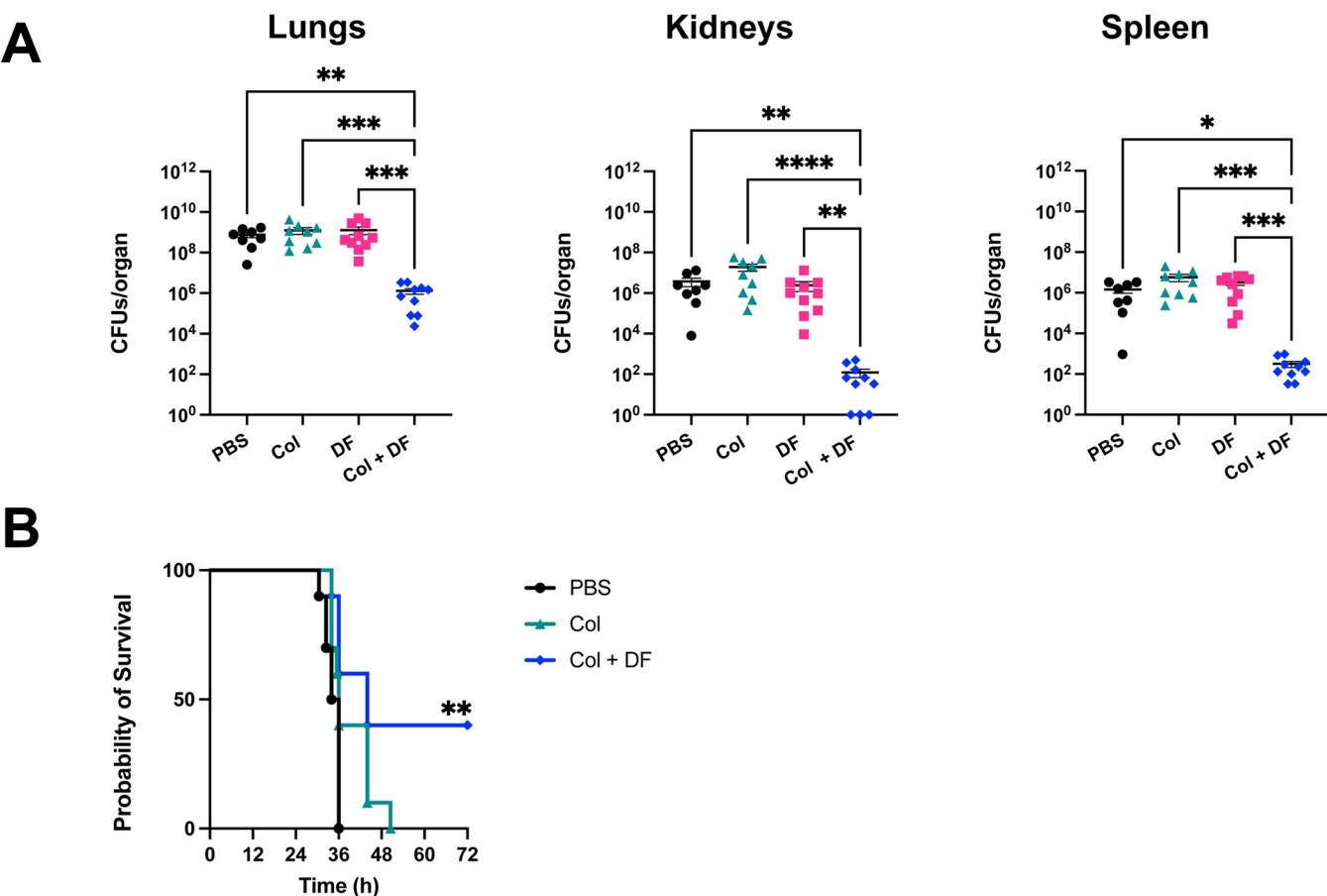

**Fig 7. ARC6851 is attenuated in an acute murine pneumonia model under colistin and diclofenac combination treatment. (A)** C57BL/6 mice were infected with ~5 × 10$^7$ CFU of mid-exponential ARC6851. At 4 hpi, mice were IP injected with 1.25 mg/kg diclofenac. At 4, 12, and 20 hpi, mice were IP injected with 2.5 mg/kg colistin. At 24 h post-infection, lungs, kidneys, and spleens were harvested, and bacterial load was determined. Each symbol represents an individual mouse, and the horizontal bar represents the Mean with SEM. Data collected from two independent experiments. **(B)** Survival rate of C57BL/6 mice (n = 10 per group) following infection with ARC6851 for the different therapies shown in **A**. *$P<0.05$, **$P<0.01$, ***$P<0.001$, ****$P<0.0001$, Kruskal-Wallis test. Col (colistin), DF (diclofenac).

parental wild-type strain (**S9B, S9C and S9D Fig**). Unlike in other bacterial species, deletion of the *pilA* gene did not affect bacterial burden in the respiratory model (**S10 Fig**). However, treatment with colistin alone was sufficient to reduce the burden of the Δ*pilA* strain in lung, kidney and spleen to a similar extent as observed with colistin plus diclofenac treatment (**Fig 8**). These results indicate that the downregulation of type IV pili is key for the synergistic activity of these drugs *in vivo*.

## Discussion

Colistin, despite its nephrotoxicity and neurotoxicity, has been used as a last resort antibiotic to treat carbapenem-resistant *A. baumannii* infections. The increasing use of colistin has resulted in the emergence of colistin-resistant strains, and therefore, alternative therapies to address carbapenem- and colistin-resistant *A. baumannii* infections are needed. Combining and repurposing existing drugs as potential therapies offers numerous advantages over traditional *de novo* drug discovery approaches, such as reduced development costs and expedited treatment implementation. In this study, we uncovered the synergistic effect between colistin

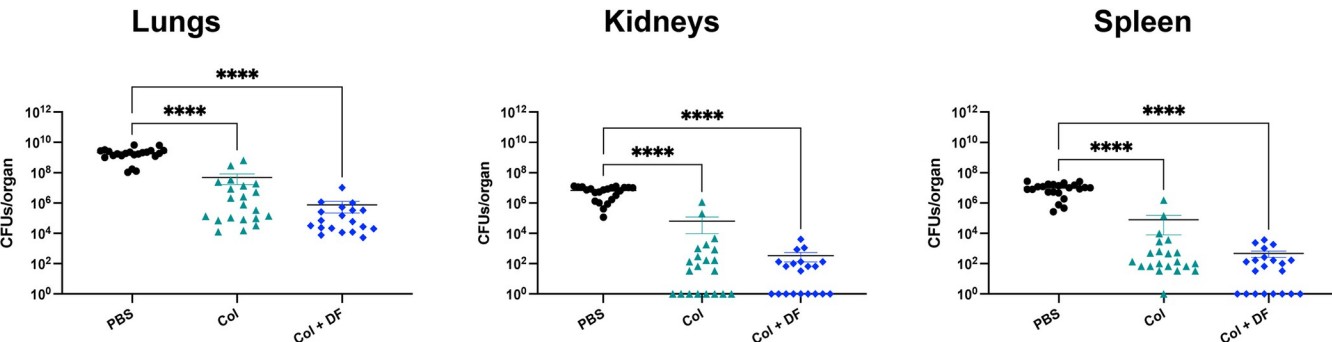

**Fig 8. ARC6851 Δ*pilA* is sensitive to colistin treatment in an acute murine pneumonia model.** C57BL/6 mice were infected with ~5 × 10$^7$ CFU of mid-exponential ARC6851 Δ*pilA*. At 4 hpi, mice were IP injected with 1.25 mg/kg diclofenac or PBS. At 4, 12, and 20 hpi, mice were IP injected with 2.5 mg/kg colistin or PBS. At 24 h post-infection, lungs, kidneys, and spleens were harvested, and bacterial load was determined. Each symbol represents an individual mouse, and the horizontal bar represents the Mean with SEM. Data collected from two independent experiments. ****$P < 0.0001$, Kruskal-Wallis test. Col (colistin), DF (diclofenac).

and diclofenac, exhibiting antibacterial activity against carbapenem- and colistin-resistant *A. baumannii*.

*A. baumannii* responds to antibiotics by differentially regulating the phenylacetic acid catabolism. Diclofenac is a non-metabolizable PAA-derivative classified as a non-steroidal anti-inflammatory drug (NSAID) approved by the U.S. FDA. Previous studies have shown that diclofenac has antibacterial activity against both Gram-negative and Gram-positive pathogens. Specifically, it has been reported that diclofenac has anti-listerial activity *in vivo* [57], is effective against *Escherichia coli* in an *in vivo* urinary model [33,34], and eliminates *Mycobacterium tuberculosis* in an *in vivo* model when used as monotherapy or in combination with streptomycin [32]. Moreover, diclofenac displayed antibacterial activity against the Gram-positive methicillin-resistant *Staphylococcus aureus* in combination with β-lactams [35]. However, we found that diclofenac did not have an effect against *A. baumannii*, *E. faecalis*, *P. aeruginosa* or *K. pneumoniae* by itself. In our study, we found that diclofenac enhanced the bactericidal activity of colistin against MDR-clinical *A. baumannii* isolates, leading to a decrease in the MIC values. Moreover, the combination treatment of colistin plus diclofenac decreased the MICs in *K. pneumoniae*, *E. faecalis* and *P. aeruginosa*, therefore expanding the potential of diclofenac to target a broader spectrum of Gram-negative pathogens.

The results of this work are summarized in **Fig 9**. To investigate the synergistic mechanism of colistin in combination with diclofenac, we performed transcriptomics and proteomics analyses. Our data showed no transcriptional changes in response to treatment with 1 μg/ml colistin. Under treatment with 100 μM diclofenac alone, we observed the upregulation of just 4 genes involved in lipid transport and metabolism, including *fadB* (acyl-CoA dehydrogenase) and *fahA* (involved in breakdown of aromatic amino acids). Remarkably, the combination of both drugs significantly changed the expression of ~130 genes. These include the upregulation of genes primarily involved in oxidative stress such as oxidoreductases, monooxygenases, and poroxiredoxin [58]. Additionally, we observed significant induction of sulfate transport and taurine metabolism pathways. This is also indicative of oxidative stress as these pathways are induced by hydrogen peroxide treatment in *A. baumannii* [58]. In fact, *Acinetobacter* was more suceptible to H$_2$O$_2$ treatment in the presence of colistin plus diclofenac and addition of the ROS scavenger DMSO rescued bacterial growth in the presence of colistin plus diclofenac, suggesting that oxidative stress is one of the ways in which both drugs synergistically kill *A. baumannii*. Moreover, during the revision of this manuscript and in agreement with our

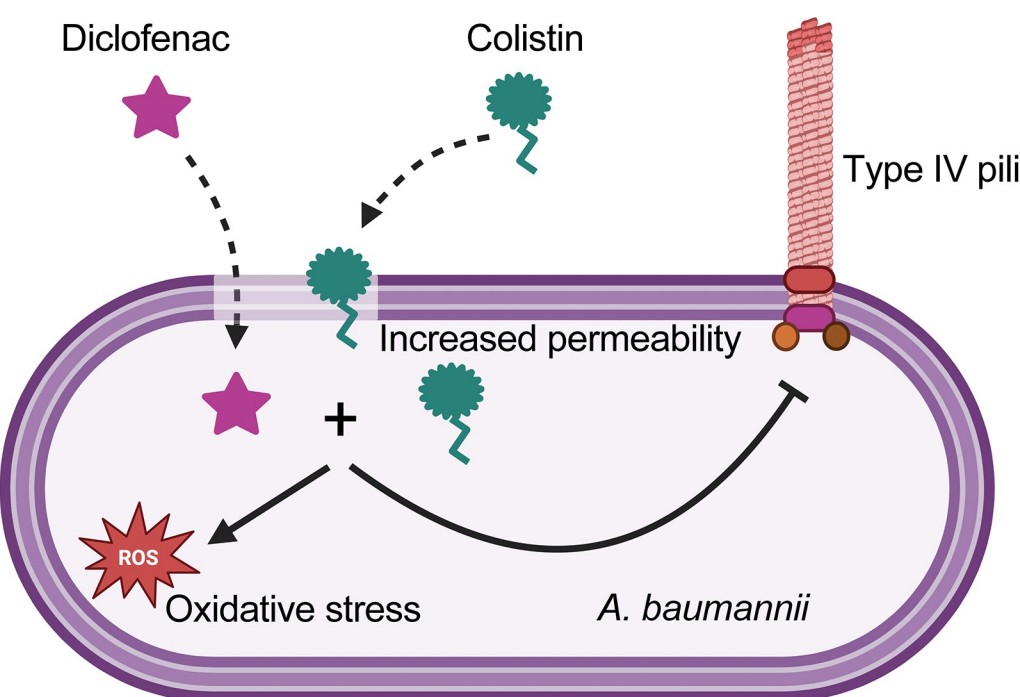

**Fig 9. Current model for the mechanism of colistin and diclofenac.** Created in Biorender.

results, Qingxia Fu et al. [59], reported that ROS production under colistin and diclofenac treatment was found to be significantly increased compared to both the monotherapy and blank treatment groups.

Type IV pili is crucial for virulence in various pathogens [52–56]. Type IV pili modulates twitching motility [60–63] and is also involved in bacterial self-organization in microcolonies and biofilms [64,65]. Prior research documented the inhibition of type IV pili-mediated biofilm in *A. baumannii* by phenothiazine compounds, a class of heterocyclic compound [66]. Remarkably, the combination treatment of colistin plus diclofenac leads to the drastic downregulation of type IV pili. Despite being as virulent as the wild-type strain in an acute murine model, the *pilA* deletion mutant strain became sensitive to colistin in the absence of diclofenac *in vivo*, indicating that the downregulation of type IV pili is key for the synergistic activity of these drugs during infection. It is tempting to speculate that repression of type IV pili by the combined therapy disrupts the formation of microcolonies or biofilm-like structures that protect the bacteria from the action of colistin. In *A. baumannii* type IV pili is generally glycosylated with negatively charged sugars [67]. It is also possible that these negatively charged sugars sequester the positively charged colistin, thereby reducing the actual concentration of colistin in the bacterial microenvironment. Further research will be necessary to fully elucidate the anti-virulence mechanism through which colistin and diclofenac are active at much lower concentrations *in vivo* than *in vitro*. The combination of colistin and diclofenac also reduced the growth of *K. pneumoniae*, *E. faecalis*, and *P. aeruginosa in vitro*. Further testing is needed to determine if the antivirulence effect observed in *A. baumannii* in mice also applies to these pathogens. This is particularly intriguing given that *K. pneumoniae* does not encode a type IV pilus. If colistin and diclofenac demonstrate an antivirulence effect in *K. pneumoniae*, it would suggest that these drugs target a different virulence factor in this bacterium. For instance, they

might affect the type II secretion system (T2SS), which is structurally related to the type IV pilus.

Both colisitin and diclofenac cause nephrotoxicity in both, human and mice [68–76]. The maximum concentration ($C_{max}$) of colistin in serum during treatment in humans ranges from 3.5 to 33.33 mg/L [77]. In mice, similar $C_{max}$ values are observed with administration of ~20 mg/kg every 8 h [50]. Regarding diclofenac, human therapeutic treatment results in a serum $C_{max}$ ranging from 2.6 to 21.5 mg/L in [78]. In mice, a single 60 mg/kg dose resulted $C_{max}$ of 264 mg/L [79]. In our experiments we achieved a drastic reduction in bacterial burden and 40% increase in survival employing doses of colistin (2.5 mg/kg) and diclofenac (1.25 mg/kg) much lower than what is typically required for human treatment. If these lower doses were equally efficient in humans, colistin and diclofenac could be employed against *A. baumannii* with minimal toxicity. Although the immediate use of diclofenac together with colistin is not precluded, it may be possible to modify the molecule to improve its antibacterial activity and reduce its side effects. Leveraging the well-understood interaction between diclofenac and COX-2, there appears to be a significant opportunity to synthesize a series of diclofenac-derivatives and select for compounds retaining their synergistic effect with colistin specifically tailored for its bacterial target but no longer able to inhibit COX-2 in the host.

## Materials and methods

### Ethics statement

All animal experiments were approved by the Washington University Animal Care and Use Committee, and we have complied with all relevant ethical regulations (Approval number 23–0071).

### Study design

The aim of this study was to assess the efficacy of combining colistin with diclofenac against multi-drug resistant (MDR) *Acinetobacter baumannii*. We proposed that FDA-approved phenylacetic acid (PAA) derivatives, like diclofenac, in combination with colistin might disrupt the bacterial response to antibiotics. We treated a highly colistin-resistant *A. baumannii* strain, ARC6851, with colistin in combination with various PAA derivatives *in vitro*. Our results indicate that colistin combined with diclofenac effectively eradicates ARC6851 and other *A. baumannii* isolates. Additionally, *Klebsiella pneumoniae* and *Enterobacter cloacae* isolates were susceptible to this combination treatment. To elucidate the molecular mechanism underlying the synergistic effect of colistin and diclofenac, we conducted permeability assays. Our findings revealed that colistin induces permeability in ARC6851, whereas diclofenac does not impact permeability. Furthermore, whole-cell transcriptomics and proteomics showed that the combination of colistin and diclofenac downregulates type IV pili and upregulates oxidative stress genes. Additionally, efflux pumps were induced by the combination treatment. Encouraged by the promising *in vitro* results, we evaluated the combination treatment for *A. baumannii* pneumonia using an acute murine infection model. This assessment demonstrated a reduction in colony-forming units (CFU) in the lungs, kidneys, and spleens. Sample sizes were determined based on prior studies [36,48], with the specific number of biological and technical replicates outlined in the figure legend for each experiment. Mice within the same litter were randomly assigned to either the control or treatment group. No animals were excluded from the experiments. A limitation of the study is that it only included female mice. It was previously observed that female mice were more susceptible to the acute pneumonia infection with *A. baumannii* that male mice [80].

### Bacterial strains and growth conditions

Bacterial strains and plasmids used in this study are listed in S12 and S13 Tables. Unless otherwise noted, strains were grown in lysogeny broth (LB) liquid medium at 37˚C with shaking (200 rpm). For cultures used for phenotypic assays, strains were struck out on solid LB-agar plates, and overnight cultures were always grown without antibiotics.

### Hoechst H33342 permeability assays

Permeability assays were performed as previously described [39, 40]. Briefly, wild-type ARC6851 cultures were grown overnight in LB. Bacterial cells were washed twice with PBS and normalized to an $OD_{600}$ of 1.0. Bacteria were then mixed with $MgSO_4$ (1 mM final concentration), colistin (128 μg/ml, 64 μg/ml, 32 μg/ml, 16 μg/ml, 8 μg/ml, 8 μg/ml, 1 μg/ml), diclofenac (100 μM) and Hoescht 33342 dye (1.25 μM final concentration). Samples were prepared in 200 μL final volume in technical sextuples and were added to a 96-well black, clear-bottomed plate. Fluorescence was monitored with excitation and emission wavelengths of 361 and 497, respectively, over the course of 2 h.

### N-phenyl-1-naphthylamine (NPN) Uptake Assay

Wild-type ARC6851 cultures were grown overnight in LB. Bacterial cells were washed twice with PBS and normalized to an $OD_{600}$ of 1.0 in 5mM Sodium HEPES, pH 7.2. Bacteria were then mixed with 128 μg/ml colistin, 100 μM diclofenac and NPN (10 μM final concentration) with or without the efflux pump inhibitor Carbonyl cyanide-m-chlorophenylhydrazone (CCCP). Samples were prepared in 200 μL final volume in technical sextuples and were added to a 96-well black, clear-bottom plate. Fluorescence was monitored with excitation and emission wavelengths of 350 and 420, respectively.

### MIC determinations

Antimicrobial susceptibility was determined using the 2-fold broth dilution microtiter assay as previously described [81,82]. Briefly, overnight cultures grown in LB were subcultured for 3 hours and inoculated at $OD_{600}$ of 0.01 in a 96-well microtiter plate containing 2-fold dilutions of the appropriate antibiotics. MICs were determined using $OD_{600}$ after plates were incubated at 37˚C under shaking conditions for 16 h. MICs were defined as <10% growth compared to the non-treated controls. Experiments were performed at least 3 independent times.

### Growth curve assays

Growth curves were performed in sterile, round-bottom, polystyrene, 96-well plates (Corning 3788). Overnight cultures were subcultured and incubated at 37˚C and 200 rpm for 3 h to mid-exponential phase. Cultures were diluted in fresh medium and inoculated at a final OD600 of 0.01 and a final volume of 150 μL. Colistin, diclofenac, DMSO, Polimixin B Nonapeptide (PBNP) and other compounds were added as indicated. Plates were incubated at 37˚C in shaking conditions for 16 h in a BioTek microplate reader, with OD600 values measured at 30 min intervals.

### Checkerboard broth microdilution assays

An overnight culture of ARC6851 grown in LB was subcultured for 3 hours. Checkerboard analyses were conducted with a clear, flat-bottom 96-well assay plate containing twofold dilutions of colistin and diclofenac in LB in an $8 \times 12$ dose-point matrix and ARC6851 culture was inoculated at $OD_{600}$ of 0.01 in the 96-well microtiter plate. The plates were incubated at 37˚C with shaking overnight for 16h then the absorbance at $OD_{600}$ was measured. MIC of the

combination colistin plus diclofenac was defined as <10% growth compared to the non-treated controls. Experiments were performed at least 3 independent times.

### $H_2O_2$ killing assays

$H_2O_2$ killing assays were performed as previously described [58]. Briefly, ARC6851overnight cultures were subcultured at $OD_{600}$ 0.05 in 10 mL LB plus DMSO, LB plus colistin (1 μg/ml) plus DMSO, LB plus diclofenac (100 μM), or LB plus colistin (1 μg/ml) plus diclofenac (100 μM) and incubated at 37°C and 200 rpm for 3 h to mid-exponential phase. Cultures were normalized to 0.3 $OD_{600}$, and 1 mL was aliquoted into 14 mL polypropylene round-bottom tubes. Cultures were treated with 0 or 5 mM $H_2O_2$ (Supelco HX0635-3) plus treatments of DMSO, colistin (1 μg/ml) plus DMSO, diclofenac (100 μM), colistin (1 μg/ml) plus diclofenac (100 μM) for 2 h at 37°C and 200 rpm. Survival was measured by serial dilution and quantification of recoverable CFUs. Experiments were performed at least five independent times.

### Culture preparation for RNA sequencing

ARC6851 was grown overnight in LB before being diluted into 10mL of LB, LB plus diclofenac (100 μM), LB plus colistin (1 μg/ml), or LB plus diclofenac and colistin and grown for 2 h at 37°C with shaking. Five individual overnight and 10mL culture biological replicates were prepared for each condition. Cultures were normalized, and the equivalent of 1 mL $OD_{600}$ 1.0 was pelleted quickly at 6.2 *x g*, treated with RNAprotect (Qiagen) for 5 minutes at room temperature, and pelleted again. Pellets were flash frozen and stored at -80°C until extraction. For RNA extractions, pellets were thawed on ice, resuspended in 600μL Trizol (Invitrogen) with 4 μl of glycogen at 5 mg/ml and lysed via bead-beating. Samples were pelleted, and supernatants were treated with chloroform. RNA was extracted from the aqueous phase using the RNeasy Mini Kit (Qiagen), and RNA quality was checked by agarose gel electrophoresis and $A_{260}/A_{280}$ measurements. RNA was stored at -80°C with SUPERase-IN RNase inhibitor (Life Technologies) until library preparation.

### RNA sequencing and analysis

RNA sequencing (RNA-Seq) was performed as previously described [83]. Briefly, 400 ng of total RNA from each sample was used for generating cDNA libraries following the RNAtag-Seq protocol. PCR amplified cDNA libraries were sequenced on an Illumina NextSeq500, generating a high sequencing depth of ~7.5 million reads per sample. Raw reads were demultiplexed by 5' and 3' indices, trimmed to 59 base pairs, and quality filtered (96% sequence quality>Q14). Filtered reads were mapped to the corresponding reference genome using bowtie2 with the-very-sensitive option (-D 20 –R 3 –N 0 –L 20 –i S, 1, 0.50). Mapped reads were aggregated by feature Count, and differential expression was calculated with DESeq2 [84, 85]. In each pair-wise differential expression comparison, significant differential expression is filtered based on two criteria: |log2foldchange| > 1 and adjusted p-value (padj) <0.05. All differential expression comparisons were made between Arc6851 in LB and either ARC6851 plus Colistin, ARC6851 plus Diclofenac, or ARC6851 plus colistin plus diclofenac. The reproducibility of the transcriptomic data was confirmed by an overall high Spearman correlation across biological replicates (R > 0.95).

### Sample preparation for Proteomic analysis

Proteomic cultures were prepared under the same conditions as RNAseq. Briefly, ARC6851 was grown overnight in LB before being diluted into 10mL of LB, LB + diclofenac (0.1 mM),

LB + colistin 1 μg/ml) or LB + diclofenac + colistin and grown for 2 h at 37°C with shaking. Five individual overnight and 10mL culture biological replicates were prepared for each condition. Whole cells were harvested by pelleting the 10 mL cultures at 4°C, washing with ice-cold phosphate-buffered saline (PBS), and pelleting again before resuspending them in 4% SDS, 100mM Tris pH 8.5. Resuspended protein samples were solubilized by boiling them for 10 min at 95°C. The protein concentrations were then assessed by bicinchoninic acid protein assays (Thermo Fisher Scientific) and 200μg of each biological replicate prepared for digestion using Micro S-traps (Protifi, USA) according to the manufacturer's instructions. Briefly, samples were reduced with 10mM DTT for 20 minutes at 95°C and then alkylated with 50mM IAA in the dark for 1 hour. Samples were acidified to 1.2% phosphoric acid and diluted with seven volumes of S-trap wash buffer (90% methanol, 100mM Tetraethylammonium bromide pH 7.1) before being loaded onto S-traps and washed 3 times with S-trap wash buffer. Samples were then digested with 4 μg of Trypsin/lys-C (Promega) overnight at 37°C before being collected by centrifugation with washes of 100mM Tetraethylammonium bromide, followed by 0.2% formic acid followed by 0.2% formic acid / 50% acetonitrile. Samples were dried down and further cleaned up using $C_{18}$ Stage [86, 87] tips to ensure the removal of any particulate matter.

## Reverse phase Liquid chromatography–mass spectrometry

$C_{18}$ enriched proteome samples were re-suspended in Buffer A* (2% acetonitrile, 0.01% trifluoroacetic acid) and separated using a two-column chromatography setup composed of a PepMap100 $C_{18}$ 20-mm by 75-μm trap (Thermo Fisher Scientific) and a PepMap $C_{18}$ 500-mm by 75-μm analytical column (Thermo Fisher Scientific) using a Dionex Ultimate 3000 UPLC (Thermo Fisher Scientific). Samples were concentrated onto the trap column at 5 μl/minutes for 6 minutes with Buffer A (0.1% formic acid, 2% DMSO) and then infused into an Orbitrap 480 (Thermo Fisher Scientific) at 300 nl/minute via the analytical columns. Peptides were separated by altering the buffer composition from 3% Buffer B (0.1% formic acid, 77.9% acetonitrile, 2% DMSO) to 28% B over 70 minutes, then from 23% B to 40% B over 4 minutes and then from 40% B to 80% B over 3 minutes. The composition was held at 80% B for 2 minutes before being returned to 3% B for 10 minutes. The Orbitrap 480 Mass Spectrometer was operated in a data-dependent mode automatically switching between the acquisition of a single Orbitrap MS scan (300–160 m/z, maximal injection time of 25 ms, an Automated Gain Control (AGC) set to a maximum of 300% and a resolution of 120k) and 3 seconds of Orbitrap MS/MS HCD scans of precursors (Stepped NCE of 28;32;40%, a maximal injection time of 55 ms, a AGC of 300% and a resolution of 30k).

## Proteomic data analysis

Identification and LFQ analysis were accomplished using MaxQuant (v2.2.0.0) [88] using the ARC6851 Proteome (NCBI accession: GCA_025677625.1 / ASM2567762v1) with Carbamidomethyl (C) allowed as a fixed modification and Acetyl (Protein N-term) as well as Oxidation (M) allowed as variable modifications with the LFQ and "Match Between Run" options enabled. The resulting data files were processed using Perseus (version 1.6.0.7) [89] with missing values imputed based on the total observed protein intensities with a range of 0.3 σ and a downshift of 1.8 σ. Statistical analysis was undertaken in Perseus using two-tailed unpaired T-tests and ANOVAs. Matching of protein homologs between the strain ATCC17978 (Uniprot accession: UP000072389) and UPAB1 Proteome (NCBI GCF_006843645.1 / ASM684364v1) was undertaken using the proteome comparison tool within PATRIC, the bacterial bioinformatics database and analysis resource [90].

## Murine pneumonia model of infection

The murine pneumonia infections were performed as previously described for *A. baumannii* [36,48]. Briefly, overnight cultures of bacteria were subcultured into 100 mL at $OD_{600}$ 0.05 and grown at 37˚C and shaking for 3 h to mid-exponential growth. Cultures were washed two times and resuspended in PBS. 6- to 8-week-old female C57BL/6 mice (Charles River Laboratories, Wilmington, MA, USA) were anesthetized by inhalation of 4% isoflurane, and then intranasally inoculated with $5 \times 10^7$ CFU of resuspended bacteria. At 4, 12, and 20 h post-infection, mice in the colistin and colistin plus diclofenac treatment groups were intraperitoneally (IP) injected with 2.5 mg/kg colistin sulfate salt in 100 µL PBS. Colistin treatment every 8 h was based on previous studies and was required due to the limited half-life of the compound in mouse serum [49–51]. At 4 h post-infection, mice in the diclofenac and colistin plus diclofenac treatment groups were IP injected with 1.25 mg/kg diclofenac sodium salt in 100 µL PBS, with the one timepoint in 24 h based on previous studies [91,92]. Groups not receiving a treatment(s) at 4, 12, and 20 h timepoints, were IP injected with a mock treatment of 100 µL PBS. At 24 h post-infection, the mice were sacrificed, and the lungs, kidneys, and spleens were aseptically removed. The bacterial load present in each tissue was determined by homogenizing each organ in PBS and plating serial dilutions on LB agar supplemented with chloramphenicol. Alternatively, survival rates were determined over a 72-hour period post infection.

## Statistical analyses

All statistical analyses were performed using GraphPad Prism version 10.

## Supporting information

**S1 Fig. Diclofenac inhibits *P. aeruginosa* growth in the presence of colistin.** Representative growth curves of *P. aeruginosa* 358800, 369569, and 409957 strains in LB containing either the solvent control DMSO or 100 µM diclofenac (DF) (left panel) and LB containing 16 µg/ml, 2 µg/ml, or 1 µg/ml colistin respectively in the presence of DMSO or 100 µM diclofenac (right panel). ****$P<0.0001$, unpaired *t* tests at 16 h for Col + DF 100 compared to DMSO control. Col (colistin), DF (diclofenac).
(TIF)

**S2 Fig. Colistin, but not diclofenac, increases membrane permeability of ARC6851. (A)** Representative membrane permeability assay measuring uptake of Hoescht 33342 fluorescent dye. Bacterial cells were suspended in PBS supplemented with increasing concentrations of colistin (Col 1 µg/ml, 8 µg/ml, 16 µg/ml, 32 µg/ml, 64 µg/ml, or 128 µg/ml) in combination with the solvent control DMSO or 100 µM DF. Relative fluorescent units (RFU) were monitored over the course of 2 hours. **(B)** Cell membrane permeabilization assay measuring NPN uptake with or without efflux pump inhibitor CCCP. ****$P<0.0001$ (One-way ANOVA with Tukey's test for multiple comparisons, in A results were compared to the control group treated with DMSO). Col (colistin), DF (diclofenac).
(TIF)

**S3 Fig. (A) Polymyxin B nanopetide and diclofenac do not affect ARC6851 growth.** Representative growth curves of ARC6851 in LB containing increasing concentrations of Polimixin B nanopeptide (PBNP) with or without 100 µM diclofenac. **(B) Diclofenac does not affect ARC6851 growth at concentrations up to 4,000 µM.** Representative growth curves of ARC6851 in LB containing increasing concentrations of diclofenac (100 µM, 200 µM, 300 µM, 400 µM, 500 µM, 1,000 µM, 2,000 µM, 3,000 µM, or 4000 µM). DF (diclofenac).
(TIF)

**S4 Fig. Colistin at 1 μg/ml does not affect ARC6851 growth.** Representative growth curves of ARC6851 in LB containing either the solvent control DMSO or 1 μg/ml colistin. Col (colistin).
(TIF)

**S5 Fig. Lipid metabolism and homeostasis genes upregulated under diclofenac treatment alone.** Volcano plot of Supplemental data set showing differentially expressed genes in ARC6851 in diclofenac treatment vs DMSO control. DF (diclofenac).
(TIF)

**S6 Fig. Functional analysis of colistin and diclofenac treatment in ARC6851. (A)** Number of differentially regulated genes in ARC6851 when treated with 1μg/ml colistin, 100 μM diclofenac, or in combination. **(B)** Donut charts representing the functional classifications of differentially regulated genes of the combination treatment against DMSO. Classifications were determined based on the designated clusters of orthologous groups (COGs) using eggNOG Mapper. Col (colistin), DF (diclofenac).
(TIF)

**S7 Fig. Type IV pili decreases in ARC6851 under colistin and diclofenac combination treatment. (A)** Relative gene expression of *pilA* in ARC6851 grown in LB + DMSO, LB+ colistin (1 μg/ml) + DMSO, LB + diclofenac (100 μM), or LB + colistin (1 μg/ml) + diclofenac (100 μM) as determined by qRT-PCR. Dotted lines represent 2-fold change. **(B)** Transmission electron microscopy of ARC6851 WT in LB plus DMSO and LB plus colistin (1 μg/ml) plus diclofenac (100 μM). Scale bar of 500 nm is shown. **(C)** Quantification of type IV pili in ARC6851 WT was performed in 20 bacterial cells. Col (colistin), DF (diclofenac). Unpaired *t* tests *$P < 0.05$.
(TIF)

**S8 Fig. Cycloxygenase COX-2 activity is not increased under diclofenac treatment.** Lung lysates containing protease inhibitors were used to determine COX-2 activities using a COX fluorescent activity assay kit. Resorufin fluorescence can be analyzed with an excitation wavelength between 530–540 nm and an emission wavelength between 585–595 nm. DF (diclofenac).
(TIF)

**S9 Fig. Colistin and diclofenac kill ARC6851 Δ*pilA* in a synergistic manner. (A)** Transmission electron microscopy of ARC6851 Δ*pilA* in LB plus DMSO and LB plus colistin (1 μg/ml) plus diclofenac (100 μM). Scale bar of 500 nm is shown. **(B)** Δ*pilA* was screened for changes in MICs to colistin in combination with the solvent control DMSO and 100 μM diclofenac using a 2-fold broth dilution method. MIC was determined as <10% growth compared to a non-treated culture. **(C)** Representative growth curves of ARC6851 Δ*pilA* in LB containing 256 μg/ml colistin with increasing concentrations of diclofenac (25 μM, 40 μM, 50 μM, 60 μM, 75 μM, 90 μM, or 100 μM). **(D)** Representative growth curves of ARC6851 Δ*pilA* in LB containing 100 μM diclofenac with increasing concentrations of colistin (1 μg/ml, 8 μg/ml, 16 μg/ml, 32 μg/ml, 64 μg/ml, 128 μg/ml, or 256 μg/ml). *$P<0.05$, ***$P<0.001$, ****$P<0.0001$.
(TIF)

**S10 Fig. ARC6851 Δ*pilA* is not attenuated in an acute murine pneumonia model.** C57BL/6 mice were infected with ~$5 \times 10^7$ CFU of mid-exponential ARC6851 WT or Δ*pilA*. At 24 h post-infection, the lungs, kidneys, and spleens were harvested, and the bacterial load present in each tissue was determined with serial dilutions. Each symbol represents an individual

mouse, and the horizontal bar represents the Mean with SEM. Data collected from two independent experiments. $*P < 0.05$, $**P < 0.01$, Kruskal-Wallis test.
(TIF)

**S1 Table. Diclofenac affects colistin susceptibility of Gram-negative pathogens.** Gram-negative and Gram-positive bacteria were screened for changes in MICs to colistin in combination with the solvent control DMSO or 100 μM diclofenac using a 2-fold broth dilution method. MIC was determined as <10% growth compared to a non-treated culture. DF (diclofenac).
(TIF)

**S2 Table. Addition of exogenous oleic acid and linoleic acid abolishes synergy between colistin and triclosan.** ARC6851 was screened for changes in MICs to triclosan in combination with either the solvent control DMSO or 20 μg/ml of oleic acid and linoleic acid, using a 2-fold broth dilution method. MIC was determined as <10% growth compared to a non-treated culture.
(TIF)

**S3 Table. Addition of exogenous fatty acids does not abolish synergy between colistin and diclofenac.** ARC6851 was screened for changes in MICs to colistin in combination with either the solvent control DMSO or 20 μg/ml of palmitic acid, araquidonic acid, linoleic acid, oleic acid, and stearic acid using a 2-fold broth dilution method. MIC was determined as <10% growth compared to a non-treated culture. Col (colistin), DF (diclofenac).
(TIF)

**S4 Table. Diclofenac does not affect *A. baumannii* susceptibility to ciprofloxacin, erythromycin, tetracycline, vancomycin, imipenem or sulfomethoxazole/trimethoprim.** ARC6851 and UPAB1 were screened for changes in MICs to ciprofloxacin, erythromycin, tetracycline, vancomycin, imipenem, or sulfomethoxazole/trimethoprim in combination with the solvent control DMSO or 100 μM diclofenac using a 2-fold broth dilution method. MIC was determined as <10% growth compared to a non-treated culture. DF (diclofenac).
(TIF)

**S5 Table. Upregulated genes in ARC6851 in diclofenac treatment vs DMSO.**
(DOCX)

**S6 Table. Differentially expressed genes in ARC6851 in colistin + diclofenac treatment vs DMSO.**
(DOCX)

**S7 Table. Differentially expressed genes in ARC6851 in colistin + diclofenac treatment vs colistin.**
(DOCX)

**S8 Table. Differentially expressed genes in ARC6851 in colistin + diclofenac treatment vs diclofenac.**
(DOCX)

**S9 Table. Differentially expressed proteins in ARC6851 in colistin + diclofenac treatment vs DMSO.**
(DOCX)

**S10 Table. Differentially expressed proteins in ARC6851 in colistin + diclofenac treatment vs colistin.**
(DOCX)

**S11 Table. Differentially expressed proteins in ARC6851 in colistin and diclofenac treatment vs diclofenac.**
(DOCX)

**S12 Table. Strains used in this study.**
(DOCX)

**S13 Table. Plasmids used in this study.**
(DOCX)

**S14 Table. Primers used in this study.**
(DOCX)

**S1 Appendix. Supplementary material and methods.**
(DOCX)

## Acknowledgments

We thank Entasis Therapeutics for the ARC6851 isolate. We thank the Melbourne Mass Spectrometry and Proteomics Facility of The Bio21 Molecular Science and Biotechnology Institute for access to MS instrumentation. We thank David Rosen for the *Klebsiella pneumoniae* strains used in this study. We thank Michele LeRoux for the *Pseudomonas* strains used in this study. We thank Dakota Hall for technical support and the members of the Feldman lab for critical reading of the manuscript. We thank the imaging laboratory of the Molecular Microbiology Department at Washington University in St. Louis. We want to thank Wandy Beatty for her collaboration in obtaining the electron microscopy images shown in this work. We thank Lance Bottini for his diligent care of our instruments.

## Author Contributions

**Conceptualization:** Fabiana Bisaro, Clay D. Jackson-Litteken, Jenna C. McGuffey, Gisela Di Venanzio, Mario F. Feldman.

**Funding acquisition:** Juan C. Ortiz-Marquez, Tim van Opijnen, Nichollas E. Scott, Mario F. Feldman.

**Investigation:** Fabiana Bisaro, Clay D. Jackson-Litteken, Jenna C. McGuffey, Anna J. Hooppaw, Sophie Bodrog, Leila Jebeli, Manon Janet-Maitre.

**Methodology:** Fabiana Bisaro, Clay D. Jackson-Litteken, Gisela Di Venanzio.

**Project administration:** Gisela Di Venanzio.

**Supervision:** Gisela Di Venanzio, Mario F. Feldman.

**Visualization:** Fabiana Bisaro, Clay D. Jackson-Litteken, Gisela Di Venanzio, Mario F. Feldman.

**Writing – original draft:** Fabiana Bisaro, Jenna C. McGuffey.

**Writing – review & editing:** Fabiana Bisaro, Clay D. Jackson-Litteken, Leila Jebeli, Nichollas E. Scott, Gisela Di Venanzio, Mario F. Feldman.

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
