## [Decision Letter · Decision Letter 0]

26 Aug 2024

Dear Dr. Feldman,

Thank you very much for submitting your manuscript "Diclofenac sensitizes multi-drug resistant Acinetobacter baumannii to colistin." for consideration at PLOS Pathogens. As with all papers reviewed by the journal, your manuscript was reviewed by members of the editorial board and by several independent reviewers. In light of the reviews (below this email), we would like to invite the resubmission of a significantly-revised version that takes into account the reviewers' comments.

Your manuscript has been reviewed by two experts and I would now like you to revise your study in line with their feedback. Both reviewers recognized the potentially high translational impact of the study, however they also highlighted the need for further exploration of mechanisms/modes of action underlying the observed interaction between diclofenac and colistin. They also requested clarification on the mechanism(s) of colistin resistance in the strains used in the study.

Reviewer #1's comments are included here:

This manuscript describes the discovery of synergies between colistin and diclofenac (DF) on killing or inhibiting the growth of colistin-resistant Acinetobacter baumannii strains in vitro and in vivo in a murine model of acute pneumonia. This synergistic effect was observed first with A. baumannii isolates in vitro and extended to other bacteria including Klebsiella pneumoniae, Enterobacter cloacae and Pseudomonas aeruginosa. Studies with A. baumannii in the mouse model indicated similar synergistic effect of colistin and DF in reducing mortality as well as bacterial loads in lungs, kidneys and spleens. The mechanisms of this synergy were investigated by transcriptomics and proteomics, with the results suggesting a role for the bacterial type IV pilus (T4P). The pilA gene encodes the pilin for T4P. Interestingly, a pilA mutant, which is as virulent in the mouse model as the wild type, still displayed a synergistic response to colistin and DF in vitro. Yet, in the mouse model, colistin alone reduced the bacterial loads of the pilA mutant without DF. The authors suggested that the mechanisms of synergy are possibly related to the negative effect of DF on T4P formation, which in turn reduces biofilm formation or sequestration of colistin by the post-translationally glycosylated pilins.

The manuscript is well written, and the work clearly described. The discovery of synergy between colistin and DF could have significant clinical implications in the treatment of colistin-resistant infections in a variety of bacteria. This is also the first example of a potential anti-T4P compound with antivirulence properties in a bacterium other than neisserial species.

I have a few minor comments on the manuscripts listed below.

1. The data on virulence by the pilA mutant is not shown, neither is data on the effect of treatments with colistin and/or DF on mouse survival. These could be provided as part of or as supplemental to Fig. 8.

2. The horizontal lines (for median) and error bars in Figs. 6, 7 & 8 appeared misplaced or missing at times. This could be a file conversion issue as well.

3. Double check Figure legends to ensure clarity and readability.

4. While the mechanism of synergy is possibly related to T4P in A. baumannii, other bacteria such as K. pneumoniae are not known to have T4P. This point may be covered in Discussion.

We cannot make any decision about publication until we have seen the revised manuscript and your response to the reviewers' comments. Your revised manuscript is also likely to be sent to reviewers for further evaluation.

Sincerely,

Daria Van Tyne

Academic Editor

PLOS Pathogens

David Skurnik

Section Editor

PLOS Pathogens

Michael Malim

Editor-in-Chief

PLOS Pathogens

orcid.org/0000-0002-7699-2064

Your manuscript has been reviewed by two experts and I would now like you to revise your study in line with their feedback. Both reviewers recognized the potentially high translational impact of the study, however they also highlighted the need for further exploration of mechanisms/modes of action underlying the observed interaction between diclofenac and colistin. They also requested clarification on the mechanism(s) of colistin resistance in the strains used in the study.

Reviewer #1 uploaded their comments as an attachment, and they are copied over here:

This manuscript describes the discovery of synergies between colistin and diclofenac (DF) on killing or inhibiting the growth of colistin-resistant Acinetobacter baumannii strains in vitro and in vivo in a murine model of acute pneumonia. This synergistic effect was observed first with A. baumannii isolates in vitro and extended to other bacteria including Klebsiella pneumoniae, Enterobacter cloacae and Pseudomonas aeruginosa. Studies with A. baumannii in the mouse model indicated similar synergistic effect of colistin and DF in reducing mortality as well as bacterial loads in lungs, kidneys and spleens. The mechanisms of this synergy were investigated by transcriptomics and proteomics, with the results suggesting a role for the bacterial type IV pilus (T4P). The pilA gene encodes the pilin for T4P. Interestingly, a pilA mutant, which is as virulent in the mouse model as the wild type, still displayed a synergistic response to colistin and DF in vitro. Yet, in the mouse model, colistin alone reduced the bacterial loads of the pilA mutant without DF. The authors suggested that the mechanisms of synergy are possibly related to the negative effect of DF on T4P formation, which in turn reduces biofilm formation or sequestration of colistin by the post-translationally glycosylated pilins.

The manuscript is well written, and the work clearly described. The discovery of synergy between colistin and DF could have significant clinical implications in the treatment of colistin-resistant infections in a variety of bacteria. This is also the first example of a potential anti-T4P compound with antivirulence properties in a bacterium other than neisserial species.

I have a few minor comments on the manuscripts listed below.

1. The data on virulence by the pilA mutant is not shown, neither is data on the effect of treatments with colistin and/or DF on mouse survival. These could be provided as part of or as supplemental to Fig. 8.

2. The horizontal lines (for median) and error bars in Figs. 6, 7 & 8 appeared misplaced or missing at times. This could be a file conversion issue as well.

3. Double check Figure legends to ensure clarity and readability.

4. While the mechanism of synergy is possibly related to T4P in A. baumannii, other bacteria such as K. pneumoniae are not known to have T4P. This point may be covered in Discussion.

Reviewer's Responses to Questions

**Part I - Summary**

Reviewer #1: (No Response)

Reviewer #2: In this work, Bisaro and coworkers explore the therapeutic potential of diclofenac, a widely used non-steroidal anti-inflamamtory drug, againt colistin resistant of A, baumannii. Whereas diclofeniac is not toxic, combination wityh colistin exerts a toxic effect against A. baumannii, and also Pseudomonas, Enterobacter, and Klebsiella. Transcriuptomic and proteomic analysis revealed that diclofenic and colistin result in upregualtion of ROS-related systtems and downregualtion of type IV pili. In vivo work probing an intranasal pneumonia model, unveiled the efficacy of combining both drugs.

General comment.

It is urgent to develop new therapeutics against infections such as those caused by A. baumannii. The idea of repurposing drugs is not novel but there is limited evidence in the case of acute infections demonstrating that it is worth exploring in clinical trials. As such this work is timeliness with translational promise. The work design and technical approach is sensible and the experiments include the relevant controls. The in vivo work is perfectly executed. Nonetheless, it remains an open question the mechanism of action of the drugs. I hope authors will find my comments useful to strengthen this important work.

**Part II – Major Issues: Key Experiments Required for Acceptance**

Reviewer #1: (No Response)

Reviewer #2: 1. At least n the case of A. baumannii authors need to provide evidence of the mechanisms of colistin resistance of the strains. This will This information should be complemented with the MIC to other antibiotics. It will be also interesting if the authors could also confirm the mechanisms of resistant in Klebsiella and Pseudomonas. To help in the understanding of the system, authors should demonstrate the mechanism of resistance of strain ARC6851.

2. The permeability-based experiments could be improved by adding uncouplers because in the current set up authors cannot differentiate membrane permeability versus efflux pumps. Also, authors may consider using as a probe NPN; it does reflect more accurately membrane permeability than the dye used.

3. A useful control experiment is to test whether diclofenac increases the susceptibility to vancomycin.

4. To probe (or not) the membrane effect, authors should test whether diclofenac increases the interaction of colistin/polymyxin B with the surface of Acinetobacter. There are commercially available labeled compounds that can be used for these experiments.

5. To further assess the membrane effect, authors should consider testing the combination of polymyxin B nanopetide and diclofenac. The nonapeptide is normally excluded (although it does interact with the LPS and disorganizes the membrane) but if the membrane affected then it can enter the cell and kill Gram-negative pathogens. The nano peptide also increases membrane permeability.

6. Authors can test whether the mechanism of action is ROS-dependent by testing ROS inhibitors. While intriguing, the results show in Fig 6 only indicate additive effect of ROS not a mechanistic connection between the drugs and their mode of action.

7. Is it possible thatt the effect on type IV pili is ROS-dependent? This can be addressed experimentally.

8. If the decrease in type IV pili is part of the mechanisms of action then the following can be anticipated: (i) in the strain background tested by the authors, a type IV pili may already display a decrease in the MIC for colistin; and.or (ii) treating this mutant with declofenac should be toxic already, and/or (iii) treating the mutant with colistin may replace the need for diclofenac. In these experiments, authors should also test the role of ROS as indicated before.

9. Authors need to justify why they only tested female mice. It is not longer appropriate to exclude one sex in infection experiments. This reviewer does not ask to repeat the in vivo work testing mal mice but urges the authors to consider this for future work. Additionally, this should be included in the disucssion as a limitation.

**Part III – Minor Issues: Editorial and Data Presentation Modifications**

Reviewer #1: (No Response)

Reviewer #2: 1. Authors may wish to show in the firsts section of the manuscript the fact that the results are colistin specific (now line 167 of the results).

2. Authors may wish to include a cartoon depicting the main findings and the mechanism of action.

3. Please do include the relevant ethics permission reference for the animal work.

PLOS authors have the option to publish the peer review history of their article (what does this mean?). If published, this will include your full peer review and any attached files.

Reviewer #1: No

Reviewer #2: No
---

## [Editor Report · Decision Letter 1]

29 Oct 2024

Dear Dr. Feldman,

We are pleased to inform you that your manuscript 'Diclofenac sensitizes multi-drug resistant Acinetobacter baumannii to colistin.' has been provisionally accepted for publication in PLOS Pathogens.

Best regards,

Daria Van Tyne

Academic Editor

PLOS Pathogens

David Skurnik

Section Editor

PLOS Pathogens

Michael Malim

Editor-in-Chief

PLOS Pathogens

orcid.org/0000-0002-7699-2064

---

## [Editor Report · Acceptance letter]

15 Nov 2024

Dear Dr. Feldman,

We are delighted to inform you that your manuscript, "Diclofenac sensitizes multi-drug resistant Acinetobacter baumannii to colistin.," has been formally accepted for publication in PLOS Pathogens.

Best regards,

Michael Malim

Editor-in-Chief

PLOS Pathogens

orcid.org/0000-0002-7699-2064